# Control of the Epithelial-to-Mesenchymal Transition and Cancer Metastasis by Autophagy-Dependent SNAI1 Degradation

**DOI:** 10.3390/cells8020129

**Published:** 2019-02-06

**Authors:** Sahib Zada, Jin Seok Hwang, Mahmoud Ahmed, Trang Huyen Lai, Trang Minh Pham, Deok Ryong Kim

**Affiliations:** Department of Biochemistry and Convergence Medical Sciences, Institute of Health Sciences, Gyeongsang National University School of Medicine, JinJu 527-27, Korea; s.zada.qau@gmail.com (S.Z.); cloud8104@naver.com (J.S.H.); ma7moud_sha3ban@hotmail.com (M.A.); tranghuyen20493@gmail.com (T.H.L.); phamminhtrang010895@gmail.com (T.M.P.)

**Keywords:** autophagy, cancer metastasis, EMT, LC3, SNAI1

## Abstract

Autophagy, an intracellular degradation process, is essential for maintaining cell homeostasis by removing damaged organelles and proteins under various conditions of stress. In cancer, autophagy has conflicting functions. It plays a key role in protecting against cancerous transformation by maintaining genomic stability against genotoxic components, leading to cancerous transformation. It can also promote cancer cell survival by supplying minimal amounts of nutrients during cancer progression. However, the molecular mechanisms underlying how autophagy regulates the epithelial-to-mesenchymal transition (EMT) and cancer metastasis are unknown. Here, we show that starvation-induced autophagy promotes Snail (SNAI1) degradation and inhibits EMT and metastasis in cancer cells. Interestingly, SNAI1 proteins were physically associated and colocalized with LC3 and SQSTM1 in cancer cells. We also found a significant decrease in the levels of EMT and metastatic proteins under starvation conditions. Furthermore, *ATG7* knockdown inhibited autophagy-induced SNAI1 degradation in the cytoplasm, which was associated with a decrease in SNAI1 nuclear translocation. Moreover, cancer cell invasion and migration were significantly inhibited by starvation-induced autophagy. These findings suggest that autophagy-dependent SNAI1 degradation could specifically regulate EMT and cancer metastasis during tumorigenesis.

## 1. Introduction

Autophagy is a cellular process that sequesters protein aggregates and damaged intracellular organelles into double-membraned autophagosomes in the cytoplasm, where these materials are degraded and recycled by a lysosome-dependent process. This phenomenon is intrinsically adopted by cancer cells to support themselves in a metabolically deficient microenvironment and maintain cellular homeostasis [1,2,3,4]. Autophagy plays a context-dependent role in cancer occurrence and progression. It increases genomic stability by removing damaged organelles and proteins, which enhances genomic stability and prevents tumorigenesis. Autophagy also promotes the survival of cancer cells under metabolic stress and improves resistance to unfavorable conditions. Therefore, autophagy represents a powerful survival strategy for cancer cells exposed to intrinsic or extrinsic stressors [5,6,7].

Cancer metastasis is a multistep process in which cells migrate from a primary site to a distant location for colonization, and it is a major cause of cancer-related deaths [8,9,10]. Metastasis requires cancer cells to detach from a primary site, migrate through the bloodstream, invade stromal tissues, and further proliferate in secondary tissues [9,10,11,12,13]. EMT plays a crucial role in the initiation of cancer metastasis and is characterized by the loss of epithelial properties that leads to mesenchymal characteristics and thereby increases cell mobility and resistance to apoptosis [14,15,16]. However, the role of autophagy in the regulation of EMT and metastasis-related proteins during tumorigenesis is unclear, although many studies suggest possible connections to cancer progression.

SNAI1/SNAIL is a key transcription factor that controls initiation of EMT [17,18,19]. It inhibits the activity of the *E-cadherin/CDH1* gene and subsequently promotes metastasis of most cancers [20,21,22,23]. Increased levels of SNAI1 also induce the self-renewal program of cancer stem-like cells by upregulating stemness factors that cause drug resistance [24,25,26]. In addition, SNAI1 has been shown to inhibit the activity of p53, which plays a crucial role in tumor suppression [19,24,27]. These findings suggest that the inactivation of SNAI1 proteins could be a potential target for the development of cancer therapies.

MAP1LC3/LC3 is a key protein involved in autophagosome formation; it regulates autophagy through its direct interaction with SQSTM1/p62. The sequence of LC3 is evolutionarily conserved from yeast to mammals. Mutations in LC3 that abrogate its ability to bind SQSTM1 cause cytotoxicity due to the excessive accumulation of SQSTM1 [28,29,30]. LC3–SQSTM1 interactions are required for degradation of polyubiquitylated protein aggregates by autophagy [30]. However, autophagy-mediated degradation of some long-lived proteins is unaffected by knockdown of the *SQSTM1* gene [31], indicating that autophagy can degrade proteins, not only via LC3–SQSTM1 interactions but also through direct interactions with LC3. Indeed, previous studies have suggested that autophagy-dependent protein degradation might be associated with cancer progression [32,33]. However, the mechanistic basis underlying how autophagy regulates EMT and metastasis is not clear.

In this study, we show that starvation-induced autophagy causes the specific degradation of SNAI1 via LC3–SQSTM1 interactions. In addition, autophagy inhibits the translocation of SNAI1 to the nucleus as well as the migration and invasion of cancer cells, suggesting that degradation of SNAI1 by autophagy is a critical process that controls tumorigenesis. Furthermore, we suggest that targeting autophagy-dependent SNAI1 degradation is a promising strategy for the development of cancer therapies.

## 2. Materials and Methods

### 2.1. Reagents

Dulbecco’s modified Eagle’s medium (DMEM, 11995-065), Roswell Park Memorial Institute 1640 Medium (RPMI-1640 (11875-119), Hank’s buffered salt solution (HBSS, 14025-092), and fetal bovine serum (FBS; 16000-044) were purchased from Gibco and Life Technologies. Chloroquine (C6628) was purchased from Sigma-Aldrich (St. Louis, MO, USA). Rapamycin (R-5000) and bafilomycin A1 were purchased from LC Laboratories (Woburn, MA, USA). Primary antibodies against LC3A/B (12741), SNAI1 (3879), TCF8/Zeb1 (3396), N-cadherin (13116), SQSTM1 (5114), phospho-ULK1 (Ser555; 5869), phospho-ULK1 (Ser757; 14202), AMPKα (2532), AMPKα T172 (2531), mTOR (2983), and phospho-mTOR (Ser2448; 2971) were from Cell Signaling Technology (Beverly, MA, USA). MAP1LC3 (SC-376404), SQSTM1/p62 (SC-28359), vimentin (sc-6601), E-cadherin (SC-7870), α-tubulin (SC-5286), and APG7 (SC-376212) were from Santa Cruz Biotechnology (Santa Cruz, CA, USA). SNAI1 (ab 53519) was from Abcam (Cambridge, MA, USA), and β-actin (A5441) was from Sigma-Aldrich. Secondary antibodies against rabbit IgG (STAR208P) and mouse IgG (STAR117P) were purchased from Bio-Rad (Hercules, CA, USA). Secondary antibodies for immunocytochemistry (FITC and TRITC) were from Santa Cruz. Protein A/G PLUS agarose immunoprecipitation reagent (sc2003) was from Santa Cruz Biotechnology. Matrigel (Corning # 344235), propidium iodide (PI), and ProLong™ Diamond antifade mountant with DAPI (# p36966) were from Invitrogen (Carlsbad, CA, USA). G488 was purchased from Thermo Scientific (Rockford, IL, USA).

### 2.2. Cell Culture

HeLa cells were cultured in DMEM containing 10% FBS. H1299 cells were cultured in the RPMI-1640 medium containing 10% FBS. All cells were grown at 37 °C in a humidified atmosphere incubator of 95% air and 5% CO_2_.

### 2.3. Western Blot Analysis

Total proteins were extracted with a cell lysis buffer supplemented with a protease and phosphatase inhibitor cocktail (Halt^TM^ Protease and Phosphatase Inhibitor Cocktail 100×, Thermo Scientific). Protein concentrations were determined using the Pierce BCA Protein Assay Kit (Thermo Scientific). Total protein lysates (30 µg) were separated by 10% SDS-PAGE, and the target proteins were specifically detected by western blotting using the indicated antibodies. Proteins were visualized with the enhanced chemiluminescence detection reagent (Thermo Scientific). All data were normalized to β-actin levels.

### 2.4. Coimmunoprecipitation Assay

Coimmunoprecipitation was performed as described in the materials (protein G agarose, sc11243233001). Briefly, HeLa cells were treated with HBSS for 4 h. Then, cells were harvested and lysed in Pierce IP lysis buffer (Thermo Scientific) supplemented with a protease and phosphatase inhibitor cocktail. Cell lysates were incubated with anti-LC3 or -SQSTM1 antibodies overnight at 4 °C with gentle rotation. Then, 50 µL of protein G agarose was added to the antibody mixture and incubated at 4 °C for 4 h with gentle shaking. Immunoprecipitates were washed with lysis buffer three times. The bound proteins were eluted by boiling the beads in 2× SDS gel loading buffer and subjected to SDS-PAGE. Specific proteins were detected by western blotting using the indicated primary antibodies.

### 2.5. Plasmid Transfection

HeLa and H1299 cells were transfected with *ATG7* shRNA or control plasmids using Lipofectamine 3000 (Invitrogen), as described in the manufacturer’s protocol. After incubation for 24 h in fresh media, cells were starved in HBSS for 4 h, and total proteins were extracted and analyzed by western blotting as described above.

### 2.6. Immunofluorescence Staining

Cells were cultured on coverslips for 24 h and then starved in HBSS for 4 h. Cells were fixed with 4% (*w*/*v*) paraformaldehyde for 30 min and permeabilized with PBS containing 0.1% Triton X-100 for 20 min at room temperature. Cells were blocked with 5% horse serum in PBS for 1 h and then incubated with primary antibodies overnight at 4 °C. After washing with PBS, cells were incubated with FITC- or TRITC-conjugated secondary antibodies (1:50 in PBS) at room temperature for 90 min. The slides were washed twice with PBS for 5 min. Glass coverslips were mounted onto glass slides using mounting medium containing DAPI. The images were captured by confocal microscopy (FV-1000; Olympus, Tokyo).

### 2.7. Subcellular Fractionation

Cells were cultured to 70–80 % confluence and then starved in HBSS for 4 h. Cells were washed with ice-cold PBS and then scraped. After centrifugation, cells were resuspended in RSB (10 mM Tris, pH 7.4, 10 mM NaCl, and 6 mM MgCl_2_) and incubated for 10 min on ice. Cells were then incubated in RSB containing 50 µM DDT, 10 mM NaF, and 1 mM NaVO_4_ on ice for 15 min. After centrifugation, the supernatants and pellets were collected as the cytosolic and nuclear fractions, respectively. Pellets were resuspended in three volumes of buffer C (20% glycerol, 20 mM HEPES, pH 7.9, 420 mM NaCl, 1.5 mM MgCl_2_, 10 mM NaF, and 1 mM NaVO_4_) and incubated on ice for 30 min. Supernatants were collected as nuclear proteins after high-speed centrifugation.

### 2.8. Wound-Healing (Scratch) Assay

Cells were grown to confluence in a 60-mm dish. Cells were treated either with 100 nM rapamycin and HBSS to induce autophagy or with 10 nM bafilomycin A1 to inhibit autophagy. A “wound” line was scratched into the cell monolayer with a sterile 1000-μL pipette tip in three separate places. The widths of the wound areas were photographed and measured under an inverted, phase-contrast microscope (Nikon, 50× magnification) to assess cell migration at 0, 24, 48, and 72 h after scratching. Distances covered by the cells from the initial wound point were measured with a ruler in centimeters.

### 2.9. Cell Invasion Assay

A transwell insert (Corning) with an 8-μm pore size was used for the two-chamber migration assay. The upper surface of the transwell insert was coated with Matrigel (BD Biosciences(San jose, CA, USA); 50 mg/filter), and HeLa cells treated with 100 nM rapamycin, HBSS, or 10 nM bafilomycin A1 were added. Cells were seeded in serum-free medium in the upper chamber. Serum-containing medium (10% FBS) was used in the lower chamber as a chemoattractant. After 36 h, cells that had invaded the bottom chamber were fixed with 3.7% formaldehyde, permeabilized with 100% methanol, stained with PI, and photographed under a microscope. Cells were quantified by counting the number of cells in 10 randomly selected areas.

### 2.10. Statistical Analysis

Each experiment was conducted independently at least three times, and results were expressed as the mean ± standard deviation (S.D.). The differences between two groups were assessed using a two-tailed Student’s *t*-test. One-way analysis of variance statistics was used to compare the means of three groups or more, followed by a Tukey’s multiple comparison test. Values of *p* less than 0.05 and *p* less than 0.01 were considered significant.

## 3. Results

### 3.1. Autophagy Specifically Regulates SNAI1 Degradation in Cancer Cells

SNAI1 is a key regulator of EMT and cancer metastasis. Thus, the intracellular levels of SNAI1 should be specifically regulated during cancer progression. Indeed, proteasomes degrade ubiquitinated SNAI1 in cancer cells [32,33,34,35,36]. To elucidate the mechanism underlying how autophagy contributes to SNAI1 degradation in cancer cells, we examined the protein levels of SNAI1 in cancer cells after autophagy was induced by starvation or treatment with rapamycin (100 nM). In both H1299 (lung cancer, Figure 1A,B) and HeLa (cervical cancer, Figure 1C,D) cell lines, SNAI1 protein levels were significantly decreased when autophagy was induced by rapamycin or starvation in HBBS. This autophagy-dependent SNAI1 degradation was significantly decreased by the addition of 20 µM chloroquine, an autophagy inhibitor. We further verified the induction of autophagy using autophagy markers, such as LC3-II formation, SQSTM1 degradation, ULK1 activity (ULK1 is activated by phosphorylation at S555 and inhibited by phosphorylation at S757), AMPK activation by phosphorylation at T172, and mTOR activation by phosphorylation at S2448. These marker proteins were appropriately activated or repressed in an autophagy-dependent manner (Figure 1A,C).

### 3.2. Autophagy-Dependent SNAI1 Degradation Is Required for LC3–SQSTM1 Interactions

Autophagy non-selectively eliminates bulky proteins and organelles, and in some instances, this process can be selective, such as the interaction that occurs between LC3 and partner proteins that have specific amino acid sequences containing an LC3-interacting region (LIR) motif [31,37]. SQSTM1, a well-known LC3-interacting adaptor protein, is involved in the degradation of associated proteins through this selective autophagy. In the context of selective SNAI1 degradation by autophagy, we used immunoprecipitation to examine the physical interaction between SNAI1 and LC3 and/or SQSTM1 under starvation conditions. Indeed, these three proteins were coimmunoprecipitated by anti-LC3 or anti-SQSTM1 antibodies, and their interaction increased in starved cells (Figure 2A,B). The SNAI1 interaction with LC3 or SQSTM1 was also evaluated by colocalization using immunocytochemistry (Figure 2C,E) and assessed quantitatively (Figure 2D,F). SNAI1 similarly colocalized with both LC3 and SQSTM1. These results indicate that autophagy degrades SNAI1 via an LC3 and/or SQSTM1-dependent manner.

### 3.3. Autophagy-Dependent SNAI1 Degradation Is Associated with EMT and Cancer Metastasis

During EMT and cancer metastasis, the levels of proteins such as N-cadherin, Zeb1, and vimentin significantly increase while the amount of E-cadherin decreases due to SNAI1-dependent transcriptional regulation [17,18,19,20,21,22,23]. To investigate how autophagy-dependent SNAI1 degradation affects the expression of EMT and metastasis-related proteins, we induced autophagy in two different cancer cell lines (H1299 and HeLa) with HBSS for 4 h in the presence or absence of 20 µM chloroquine. In both cell types, autophagy was similarly activated by HBSS treatment. Activation of ULK1 and AMPK and degradation of SQSTM1 were observed after starvation. Moreover, the levels of SNAI1 proteins were dependent on the induction of autophagy (Figure 3A–C), and there was a large decrease in the levels of mesenchyme-related proteins (including N-cadherin, and Zeb1) upon starvation-induced autophagy (Figure 3B,C). However, vimentin was not significantly changed in H1299 cells (Figure 3B,C). The expression levels of these proteins significantly increased by inhibiting autophagy with 20 µM chloroquine. In contrast, the levels of epithelial proteins, such as E-cadherin, were highly increased by the induction of autophagy under starvation and decreased after treatment with 20 µM chloroquine in all cancer cell lines (Figure 3). We further examined the expression levels of SNAI1 using immunocytochemistry. Indeed, the intensity of SNAI1 was decreased in starved cells but increased after treatment with chloroquine (Figure 3D,E), suggesting that the intracellular levels of SNAI1 are specifically regulated by autophagy and functionally associated with the regulation of cancer progression.

### 3.4. ATG7 Knockdown Rescues Starvation-Induced SNAI1 Degradation

Because ATG7 plays a crucial role in the autophagy process, knockdown of the *ATG7* gene was shown to cause a severe defect in autophagy via LC3 lipidation with phosphatidylethanolamine [38]. To investigate how the knockdown of the *ATG7* gene affects SNAI1 degradation, we transiently transfected plasmids containing *ATG7* shRNA and control shRNA into two cancer cell lines (HeLa and H1299) and incubated the cells for 24 h. The expression levels of ATG7 in both HeLa and H1299 cells were significantly decreased after treatment with *ATG7* shRNA, and starvation-induced autophagy was subsequently found to be defective after *ATG7* knockdown (Figure 4). Furthermore, autophagy defects induced by the *ATG7* knockdown significantly inhibited the degradation of SNAI1 in both cancer cell lines (Figure 4), indicating that ATG7-dependent autophagy is crucial for controlling the protein levels of SNAI1 in cancer cells.

### 3.5. Starvation-Induced Autophagy Inhibits SNAI1 Translocation to the Nucleus

SNAI1, a transcription factor, is translocated into the nucleus where it inhibits the transcription of epithelial markers such as *E-cadherin/CDH1* and promotes metastasis of most cancers [20,21,22,23]. In this study, we examined the intracellular levels of SNAI1 in both the cytosol and nucleus when HeLa cells were starved in HBSS in the presence or absence of 20 µM chloroquine. In both the cytosol and nucleus, the levels of SNAI1 were significantly decreased after starvation, although nuclear SNAI1 levels were less affected than cytosolic SNAI1 levels (Figure 5A). Similarly, SNAI1 levels in both the cytosol and nucleus recovered after the inhibition of autophagy by chloroquine (Figure 5B,C). Furthermore, according to immunocytochemical data, the nuclear localization of SNAI1 was reduced after cells were starved and substantially increased after treatment with chloroquine (Figure 5D,E). These data demonstrate that autophagy degrades SNAI1 in the cytoplasm and consequently inhibits SNAI1 translocation to the nucleus.

### 3.6. Autophagy Inhibits Cancer Cell Migration and Invasion

As described above, the intracellular levels of SNAI1 are regulated by autophagy. To investigate whether autophagy-induced SNAI1 degradation influences the migration and invasion of cancer cells, we performed scratch assays under different conditions. Autophagy was induced by treatment with 100 nM rapamycin or HBSS (starvation), resulting in reduced wound closure activity by HeLa cancer cells. In contrast, the inhibition of autophagy with 10 nM bafilomycin A1 largely increased the wound closure activity of HeLa cells in normal media and under starvation conditions (Figure 6A,B). In these experiments, we used 10 nM bafilomycin A1 rather than 20 µM chloroquine to inhibit autophagy because chloroquine revealed high cytotoxicity in case of long-term incubation. We also examined cell invasion under the same conditions using the Matrigel chamber assay. Induction of autophagy decreased cell invasion while its inhibition with bafilomycin A1 significantly increased the invasion activity of cancer cells (Figure 6C,D). Additionally, similar results were observed in H1299 cells (data not shown). These results indicate that autophagy plays an important role in cancer migration and invasion via the possible modulation of intracellular SNAI1 levels.

## 4. Discussion

Autophagy is a process by which cellular metabolic homeostasis is maintained by delivering excess and unnecessary cytoplasmic materials to double-membraned autophagosomes, which ultimately fuse with lysosomes for the degradation of their contents [1,2,3]. Cancer cells utilize autophagy not only for their metabolic demands but also to escape stress-induced cell death [5,6,7]. Since autophagy plays multiple roles in tumor progression, it is one of the major targets of cancer therapy, although this depends on both cancer cell type and context.

According to previous works, autophagy inhibits EMT and metastasis in gastric cancer [32,39]. Autophagy impairs the migration and invasion of cancer cells by downregulating Snail and Slug [40]. Conversely, silencing autophagy-related proteins restores the mesenchymal phenotype [32,33,39]. Similarly, our study shows that starvation-induced autophagy leads to the degradation of intracellular SNAI1 and the suppression of cancer progression. In contrast, blocking autophagy with chemicals or by gene silencing increases SNAI1 protein levels and leads to normal cancer progression. Moreover, other studies have suggested that autophagy attenuates EMT and tumor metastasis by the degradation of Twist1 and SNAI1 [33,41]. In addition to cancers, autophagy-dependent SNAI1 degradation also determines hepatocyte identity by regulating the transition between EMT and MET. Indeed, stimulation of autophagy in hepatocytes strongly affects EMT by promoting SNAI1 degradation, and conversely, TGF β-induced EMT leads to autophagy impairment [42]. These all studies suggest that autophagy might play a key role in controlling the intracellular level of SNAI1, which is essential to determine cell transitions in cancer and hepatocyte differentiation.

SNAI1 acts as a main driver of EMT and metastasis [17,18,19]. It specifically inhibits the transcription of anticancer genes such as *CDH1/E-cadherin* and activates mesenchymal markers to promote the metastasis of most cancers [20,21,22,23]. Our results show that the intracellular levels of E-cadherin decrease when cancer cells are starved and are increased by the inhibition of autophagy, suggesting that autophagy-dependent SNAI1 degradation is directly associated with its downstream targets such as E-cadherin. Additionally, other EMT-related proteins including N-cadherin, and Zeb1 are likely regulated in an autophagy-dependent manner (Figure 3).

In this study, we suggest that SNAI1 directly binds to LC3 in cancer cells. LC3 plays a key role in the initiation of autophagosome formation. It also mediates autophagy by its association with specific proteins (e.g., SQSTM1) through a highly conserved protein domain, called an LIR. The LIR contains two hydrophobic amino acids (e.g., tryptophan, W; leucine, L; and phenylalanine, Y) separated by two nonspecific amino acids (X). Some proteins that possess an LIR are selectively degraded by autophagy through a direct link with LC3, whereas other ubiquitinated proteins are engulfed by an autophagosome via an indirect association with the LC3-SQSTM1 complex. Interestingly, we observed that the amino acid sequence of SNAI1 contains three putative LIR motifs (^61^WXXV^64^, ^74^WXXL^77^, and ^163^YXXL^166^), and that SNAI1 was physically associated with LC3 in a coimmunoprecipitation assay and by a colocalization analysis (Figure 2). In addition, SNAI1 was associated with SQSTM1, indicating that SNAI1 can be removed by autophagy in an SQSTM1-dependent manner. Based on these results, we suggest that intracellular SNAI1 protein levels are controlled by autophagy during cancer progression, which is consistent with results from previous studies in which autophagy inhibited cancer growth by degrading SNAI1 [32,33].

Both EMT and cancer metastasis are stimulated by an increase in SNAI1, N-cadherin, Zeb1, and vimentin, and a decrease in E-cadherin [17,18,19,20,21,22,23]. In this study, we show a similar decrease in SNAI1, N-cadherin, and Zeb1 upon starvation-induced autophagy in two cancer cell lines. Conversely, E-cadherin levels were increased by the induction of autophagy in cancer cells (Figure 3). Since three other EMT-related proteins (N-cadherin, Zeb1, and vimentin) are downstream mediators of SNAI1, they should be dependent on the induction of autophagy. Indeed, the intracellular levels of these three proteins were also regulated by autophagy, in contrast to that of SNAI1. These three proteins were inhibited by the induction of autophagy and increased in response to the inhibition of autophagy. These results agree with those of Catalano et al. [32] and other studies [42,43] in which autophagy was found to reverse EMT. The connections between EMT-related proteins and their association with autophagy are presented in Figure 7.

In conclusion, SNAI1 is selectively degraded during starvation-induced autophagy by its interaction with SQSTM1 or LC3. SNAI1 degradation inhibits cancer progression by downregulating many EMT-related proteins. As a result, autophagy-induced SNAI1 degradation in the cytoplasm inhibits SNAI1 translocation into the nucleus and blocks the transcription of many genes involved in cancer cell invasion and migration. This study suggests that autophagy could be a possible target for the development of therapeutic strategies to inhibit cancer progression and modulate the stem cell-like properties and chemoresistance of cancer cells.

## Figures and Tables

**Figure 1 cells-08-00129-f001:**
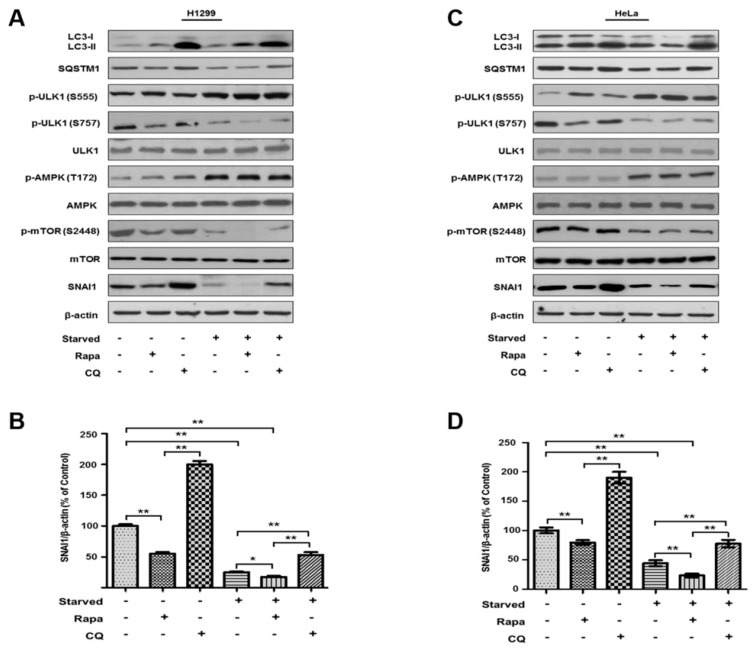
Autophagy stimulates SNAI1 degradation in cancer cells. H1299 (**A**) and HeLa (**C**) cells were treated with 100 nM rapamycin (Rapa) or starved in HBSS medium for 4 h to induce autophagy in the presence or absence of 20 µM chloroquine (CQ). After lysis, total cell extracts (30 µg) were separated by 10% or 12% SDS-PAGE and analyzed by western blotting using primary antibodies against LC3, SQSTM1, p-ULK1-S555, p-ULK1-S757, p-AMPK-T172, AMPK, p-mTOR-S2248, mTOR, and SNAI1. β-actin was used as a loading control. The levels of SNAI1 in H1299 (**B**) and HeLa cells (**D**) were quantified using NIH ImageJ software. Data represent the mean (±S.D.) of three independent experiments (* *p* < 0.05 and ** *p* < 0.01).

**Figure 2 cells-08-00129-f002:**
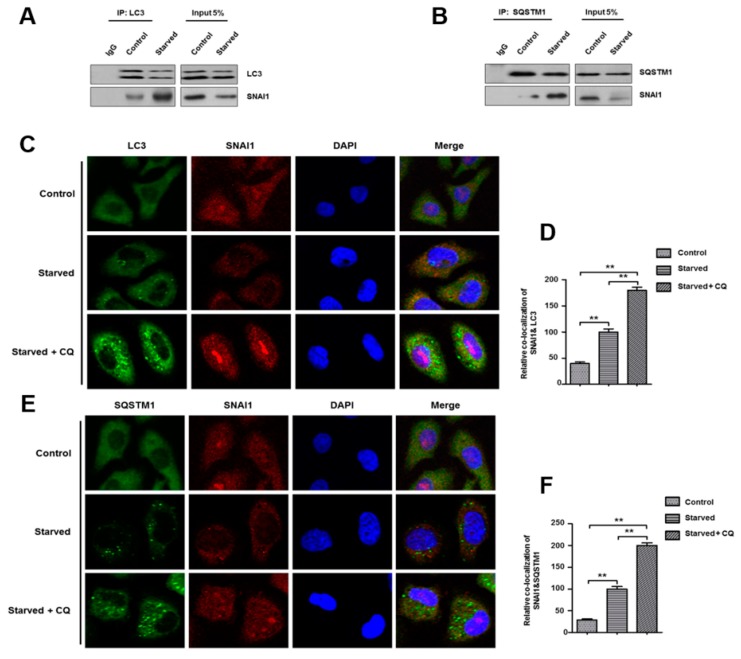
Possible interactions between SNAI1 and LC3 or SQSTM1/p62. (**A**,**B**) Coimmunoprecipitation. Control or starved HeLa cell lysates were immunoprecipitated by anti-LC3 (**A**) or anti-SQSTM1 (**B**) antibodies in 30 µL protein G agarose beads. After washing, the bound proteins were analyzed by western blotting using the indicated antibodies. Whole-cell extracts (5% input) were also assessed by western blotting. IgG indicates nonspecific mouse antibody as a negative control. (**C**,**E**) Immunocytochemistry. HeLa cells were cultured on coverslips for 24 h and then starved with HBSS for 4 h. After fixing cells with paraformaldehyde, cells were incubated for 24 h at 4 °C with rabbit polyclonal anti-SNAI1 plus mouse monoclonal anti-LC3 (**C**) or mouse monoclonal anti-SQSTM1 (**E**) antibodies. After washing, secondary antibodies (FITC-conjugated anti-mouse antibody or TRITC-conjugated anti-rabbit antibody) were applied to cells. The glass slides were mounted using a mounting medium containing DAPI (to stain nuclei), and all images were captured by confocal microscopy (Olympus FV-1000). (**D**,**F**) Colocalization of SNAI1 with LC3 (**D**) and SQSTM1 (**F**) was quantified using NIH ImageJ software. Data represent the mean (±S.D.) of three independent experiments (** *p* < 0.01).

**Figure 3 cells-08-00129-f003:**
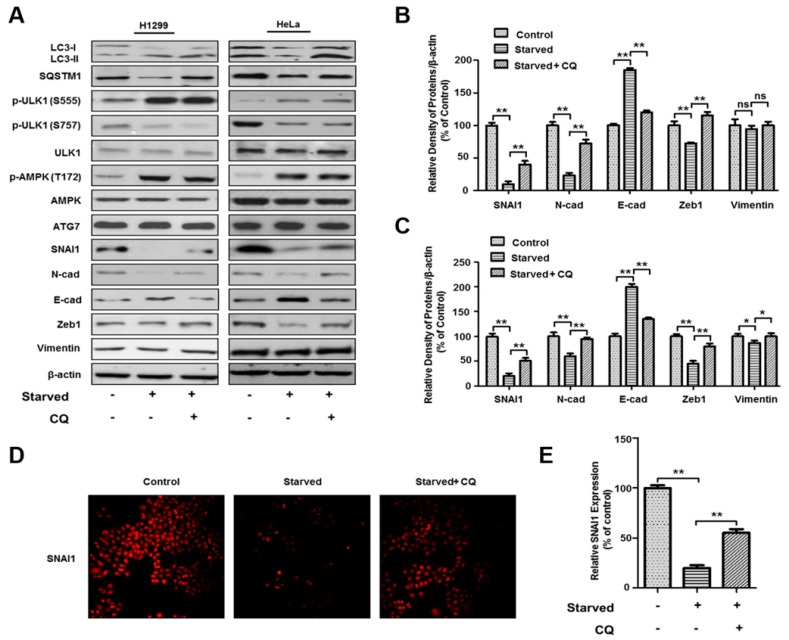
Autophagy-induced SNAI1 degradation is associated with the expression of EMT- and cancer metastasis-related proteins. (**A**) H1299 (left) and HeLa (right) cells were starved with HBSS for 4 h in the presence or absence of 20 µM chloroquine (CQ). After lysis, total-cell extracts (30 µg) were separated by 10% or 12 % SDS-PAGE and analyzed by western blotting using primary antibodies against LC3, SQSTM1, p-ULK1-S555, p-ULK1-S757, p-AMPK-T172, AMPK, p-mTOR-S2248, mTOR, SNAI1, N-cadherin, E-cadherin, Zeb1, and vimentin. β-actin was used as a loading control. (**B**,**C**) The levels of each EMT-related protein (SNAI1, N-cadherin, Zeb1, and vimentin) in H1299 (B) or HeLa (**C**) cells were quantified using NIH ImageJ software. β-actin was used as a loading control. Data represent the mean (± S.D.) of three independent experiments (* *p* < 0.05 and ** *p* < 0.01). (**C**,**D**) Immunocytochemistry of SNAI1. HeLa cells were treated with HBSS with or without 20 µM CQ for 4 h. The expression levels of SNAI1 (red) in cells were determined by immunocytochemistry (**C**). Images were captured from at least 20 different areas by florescence microscopy (Olympus BX51-DSU), and the immunofluorescence of SNAI1 was quantified using NIH ImageJ software (**D**). Data represent the mean (±S.D.) of three independent experiments (** *p* < 0.01).

**Figure 4 cells-08-00129-f004:**
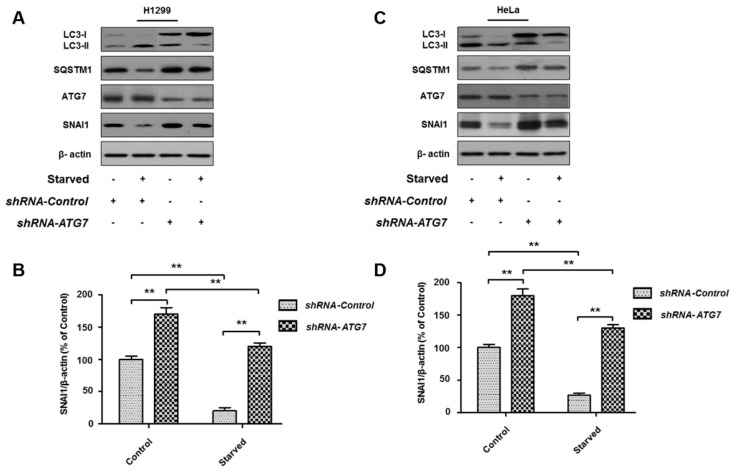
*ATG7* knockdown rescues SNAI1 degradation. H1299 (**A**) or HeLa (**C**) cells were transiently transfected with *ATG7* shRNA and control shRNA plasmids using Lipofectamine 3000 and incubated for 24 h. Cells were then starved with HBSS for 4 h. Total-cell proteins (30 µg) were separated by 10% or 12% SDS-PAGE and analyzed by western blotting using antibodies against LC3, SQSTM1, ATG7, and SNAI1. β-actin was used as a loading control. (**B**,**D**) Quantification of SNAI1. The levels of SNAI1 in H1299 cells (**B**) and HeLa cells (**D**) were quantified using NIH ImageJ software. β-actin was used as a loading control. Data represent the mean (±S.D.) of three independent experiments (** *p* < 0.01).

**Figure 5 cells-08-00129-f005:**
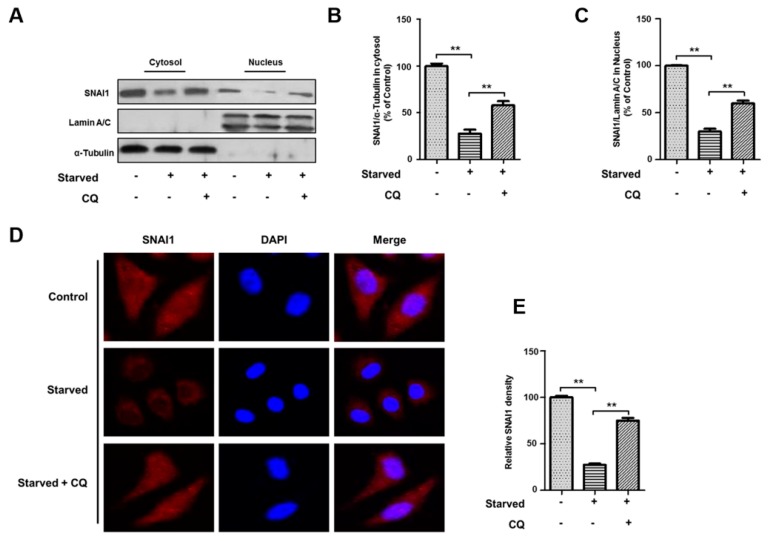
Autophagy-dependent degradation of SNAI1 in the cytosol inhibits SNAI1 translocation into the nucleus. (**A**) HeLa cells were starved in HBSS for 4 h with or without 20 µM chloroquine (CQ). Whole cell extracts were fractionated into cytosolic and nuclear extracts as described in the methods. Each fraction was analyzed by western blotting analysis. Both α-tubulin and lamin A/C were used as loading controls for the cytosolic and nuclear fractions, respectively. (**B**,**C**) Quantification of SNAI1. The levels of SNAI1 in the cytosol (**B**) and nucleus (**C**) were quantified using NIH ImageJ software. Data represent the mean (±S.D.) of three independent experiments (** *p* < 0.01). (**D**,**E**) Localization of SNAI1 in the cytosol and nucleus. HeLa cells were starved in HBSS in the presence or absence of 20 µM CQ. SNAI1 staining (**C**) in the cytosol or nucleus was determined by immunohistochemistry. Images were captured from at least 20 different areas by confocal microscopy (Olympus FV-1000), and the immunofluorescence of nuclear SNAI1 was quantified using NIH ImageJ software (**D**). Data represent the mean (±S.D.) of three independent experiments (** *p* < 0.01).

**Figure 6 cells-08-00129-f006:**
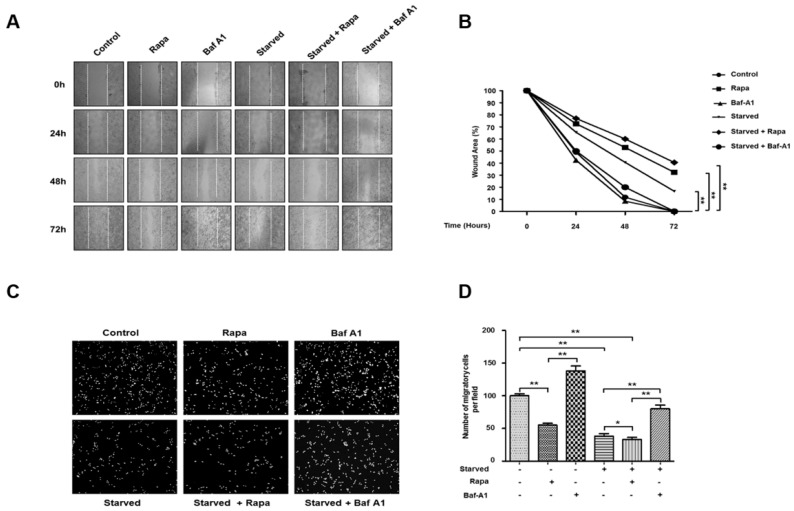
Autophagy inhibits migration of cancer cells. (**A**,**B**) HeLa cells were treated with either 100 nM rapamycin (Rapa) or HBSS to induce autophagy in the presence or absence of 10 nM bafilomycin A1 (BafA1). The migratory behavior of cancer cells under different conditions was determined by the wound-healing assay, as described in the methods, and representative images (50× magnification) indicate cell migration at the times indicated. The wound edges at 0 h are indicated by white dotted lines. Cell migration (**B**) is represented as wound closure (percentage of the average migratory distance of cells relative to the original wound edge). Data indicate the mean (±S.D.) of at least three independent experiments (* *p* < 0.05 and ** *p* < 0.01). (**C**) The Matrigel invasion chamber assay using HeLa cells. Cells that invaded the transwell insert were stained with PI and photographed. (**D**) Cells were quantified in 10 randomly selected fields under a microscope (10× objective magnification). Data represent the mean (±S.D.) of three independent experiments (** *p* < 0.01).

**Figure 7 cells-08-00129-f007:**
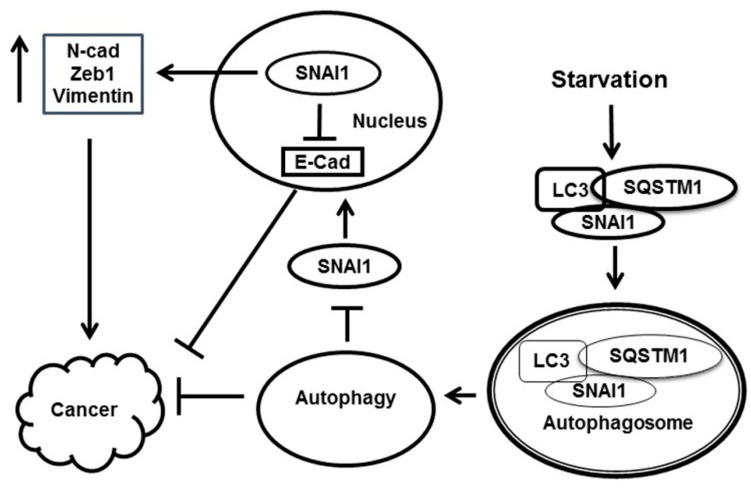
Schematic representation of autophagy-dependent SNAI1 degradation during cancer progression. Starvation-induced autophagy leads to the degradation of SNAI1 via the incorporation of LC3–SQSTM1–SNAI1 complexes into autophagosomes. Consequently, the low levels of nuclear SNAI1 cause a decrease in the levels of EMT-related proteins (N-cadherin, Zeb1, and vimentin) and an increase in E-cadherin through transcriptional deactivation. Thus, the induction of autophagy inhibits cancer progression.

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
