# Peer review of "Control of the Epithelial-to-Mesenchymal Transition and Cancer Metastasis by Autophagy-Dependent SNAI1 Degradation"

_cells, 2019, doi:10.3390/cells8020129_

Round 1

Reviewer 1 Report

I still believe that presenting co-localization in this way is not correct.

the rest of manuscript has been sufficiently improved 

Author Response

I still believe that presenting co-localization in this way is not correct. the rest of manuscript has been sufficiently improved 

(Our response) As suggested by the reviewer, we carried out co-localization imaging assays using confocal microscope (Olympus FV-1000) and replaced old images with new confocal ones in Figure 2 and 5 together with their quantification. Furthermore, we observed a similar result as shown previously.

Reviewer 2 Report

The authors have not adequately resolved the issue of determining protein co-localization. Since this is such a big part of this study the authors should at least attempt to confirm their findings with confocal microscopy, as the reader has no way of evaluating their specific software.

Author Response

The authors have not adequately resolved the issue of determining protein co-localization. Since this is such a big part of this study the authors should at least attempt to confirm their findings with confocal microscopy, as the reader has no way of evaluating their specific software.

(Our response) As suggested by the reviewer, we carried out co-localization imaging assays using confocal microscope (Olympus FV-1000) and replaced old images with new confocal ones in Figure 2 and 5 together with their quantification. Furthermore, we observed a similar result as shown previously.

This manuscript is a resubmission of an earlier submission. The following is a list of the peer review reports and author responses from that submission.

Round 1

Reviewer 1 Report

In the manuscript by Sahib et al. the authors  sustain that autophagy-mediated SNAI1 degradation suppresses EMT and cancer metastasis. Although interesting, the work lacks of novelty and some data are questionable.

major issues:

1-     I was disappointed by the lack of citation of the work by Grassi et al (Cell Death Dis. 2015 Sep 10;6:e1880. doi: 10.1038/cddis.2015.249) which already demonstrated SNAI1 autophagy-dependent degradation through p62, suggesting that autophagy is impaired during the TGFβ-induced EMT and, viceversa, induction of autophagy inhibits the TGFβ-mediated EMT. Their studies were conducted on hepatocytes but their work provides a previous suggestion of the role of autophagy and SNAIL1 in EMT accomplishment.

2-     Figure 1: LC3 levels in H1299 cell line are controversial. The levels of LC3 in control cells are very low, indicating autophagy activity, while in Rapamycin treated and starved cells the levels are higher. On the contrary, in Hela cell line LC3 levels are higher in control sample as in Rapamycin treated cells, while in starved cells and in rapa+starv cells is (as expected) lower. How do the authors explain these discrepancies? Moreover, in figure 1b, SNAI1 WB quantification shows that SNAI1 protein level is higher in starved cells than in rapa+starv cells, which looking at the WB signal seem not correct. Since the quantification is performed on 3 experiments, please provide a representative WB that better reflects the quantification data.

3-     Figure2: panels A and B the authors performed IPs with antiLC3 antibody (A) and anti-p62 antibody (B) then they show the co-IP for SNAI1 and for p62 in both cases. I would expect a co-IP for LC3 in the panel B and, eitherway, I think the authors should add the panel showing that the correct IP occurred (LC3 for panel  A and p62 for panel B)and how the starvation treatment influenced the IP.

Panels C and D: I really can’t appreciate the re-location of SNAI1 in dots, the protein seems to me diffuse all-over the cell and, although it can be appreciated its downregulation upon starvation, its co-localization with LC3 and p62 seems very overstated to me.

4-     The authors state that there was a large decrease in the levels of mesenchymal proteins including NCAD, Zeb1 and vimentin, which expression was significantly recovered inhibiting autophagy. Looking at Figure 3, the quantification of Zeb1 and Vimentin seems odd and not in line with WB images. I suggest to change the statements (also in discussion) since WB images do not reflect quantification data.

5-     ATG7 knockdown is only showed in combination with starvation treatment, for completeness please add WB of ATG7 siRNA alone. Starvation+siRNA data are convincing, but autophagy induction cannot be appreciated from LC3 and SQSTM1 protein levels, in both cell lines but particularly in H1299 cells, in which LC3 and p62 protein levels seem the same as control, while SNAI1 is downregulated. Can the authors discuss that?

6-     For figure 5 and 6 experiments, the authors focused only on HeLa cell lines, not performing further experiments on H1299. Since the two cell lines do not refer to the same cancer type, can they show if also in H1299 cells they obtained similar results and discuss that?

7-     Figure 5D is quite misleading. Even if the downregulation of SNAI1 can be appreciated, the statement that its nuclear translocation is inhibited by autophagy  cannot be corroborated based on these data. The data only show its general degradation, which is not necessarily associated with its nuclear translocation.

8-     In the discussion there are several overstatements, as described above, which must be changed. Furthermore, it would be interesting to expand the discussion with previous works that described a similar EMT/Autophagy interplay, like Gugnoni et al (Oncogene. 2017 Feb 2;36(5):667-677. doi: 10.1038/onc.2016.237.), Catalano et al (Mol Oncol. 2015 Oct;9(8):1612-25. doi: 10.1016/j.molonc.2015.04.016.), Lv et al. (Cancer Res. 2012 Jul 1;72(13):3238-50. doi: 10.1158/0008-5472.CAN-11-3832.) or others.

Some minor points are mainly on figures labels (Cq indicated when Bafilomycin is used in fig 6D, an additional “C” to mark the panels in figure 3, Immunohistochemistry instead of immunocytochemistry in paragraph 3.2) and in materials an methods where a lot of drugs and compounds concentrations miss the unit of measure.

Author Response

Reviewer1:

In the manuscript by Sahib et al. the authors  sustain that autophagy-mediated SNAI1 degradation suppresses EMT and cancer metastasis. Although interesting, the work lacks of novelty and some data are questionable.

major issues:

1. I was disappointed by the lack of citation of the work by Grassi et al (Cell Death Dis. 2015 Sep 10;6:e1880. doi: 10.1038/cddis.2015.249) which already demonstrated SNAI1 autophagy-dependent degradation through p62, suggesting that autophagy is impaired during the TGFβ-induced EMT and, viceversa, induction of autophagy inhibits the TGFβ-mediated EMT. Their studies were conducted on hepatocytes but their work provides a previous suggestion of the role of autophagy and SNAIL1 in EMT accomplishment.

Our response) Honestly we have carried our whole work without recognizing the existence of the work by Grassi et al (Cell Death Dis. 2015 Sep 10;6:e1880. doi: 10.1038/cddis.2015.249), so we missed the citation in our manuscript. In the revised manuscript, we have included the work by the Grassi et al as a reference. As mentioned by the reviewer, Grassi et al suggested that Snail proteins can be regulated by SQSTM1/p62 dependent autophagy in hepatocytes. In particular, they found that autophagy-dependent Snail level determines hepatocyte identity via balance between EMT and MET.

In this work we mainly focused on the autophagy-dependent regulation of intracellular level of Snail in cancer cells. Also we suggest that Snail degradation is mediated by the direct interaction between Snail and LC3 without involving SQSTM1/P62.  

2.    Figure 1: LC3 levels in H1299 cell line are controversial. The levels of LC3 in control cells are very low, indicating autophagy activity, while in Rapamycin treated and starved cells the levels are higher. On the contrary, in Hela cell line LC3 levels are higher in control sample as in Rapamycin treated cells, while in starved cells and in rapa+starv cells is (as expected) lower. How do the authors explain these discrepancies? Moreover, in figure 1b, SNAI1 WB quantification shows that SNAI1 protein level is higher in starved cells than in rapa+starv cells, which looking at the WB signal seem not correct. Since the quantification is performed on 3 experiments, please provide a representative WB that better reflects the quantification data.

Our response) To determine autophagy using LC3-II levels is very controversial because they are different in cell types, incubation time points, and others. So many researchers suggest that not only LC3 levels but also other indices of autophagic flux are recommended to autophagy. The autophagic index using LC3 can be determined by the relative ratio between decrease in LC3-I and increase in LC3-II indicating the conversion of LC3 I- to LC3 II. Here in H1299 cells the conversion of LC3 I- to LC3 II accordingly occurred and we also showed the level of p62 level, P-ULK1 (S555) P-ULK1 (S757), P-AMPK(T172) and P-mTOR (S2448) because only LC3 is not good indicator for autophagic flux. Furthermore, autophagy is a complex process dependent on different contexts.

Also, in many cells LC3-II level was decreased with induction of autophagy as reported in many previous studies including the following ones: Noboru Mizushima & Tamotsu Yoshimori (2007) How to Interpret LC3 Immunoblotting, Autophagy, 3:6, 542-545, DOI: 10.4161/auto.4600; Noboru Mizushima, Tamotsu Yoshimorim and Beth Levine, Methods in Mammalian Autophagy Research. Cell. 2010 February 5; 140(3): 313–326. doi:10.1016/j.cell.2010.01.028; Peidu Jiang and Noboru Mizushima, LC3- and p62-based biochemical methods for the analysis of autophagy progression in mammalian cells. Methods 75 (2015) 13–18; Ji-Seon Ahn et al Autophagy negatively regulates tumor cell proliferation through phosphorylation dependent degradation of the Notch1 intracellular domain. Oncotarget. 2016; 7:79047-79063. https://doi.org/10.18632/oncotarget.12986

In Fig 1 B SNAI1 level looks lower because it was on x-ray film dot. However, we added new results.

3.     Figure2: panels A and B the authors performed IPs with antiLC3 antibody (A) and anti-p62 antibody (B) then they show the co-IP for SNAI1 and for p62 in both cases. I would expect a co-IP for LC3 in the panel B and, either way, I think the authors should add the panel showing that the correct IP occurred (LC3 for panel  A and p62 for panel B)and how the starvation treatment influenced the IP.

Panels C and D: I really can’t appreciate the re-location of SNAI1 in dots, the protein seems to me diffuse all-over the cell and, although it can be appreciated its downregulation upon starvation, its co-localization with LC3 and p62 seems very overstated to me.

Our response)  In Figure 2 A&B, the figures have been revised as suggested by reviewer.

In Figure 2 C&D, the reviewer’s suggestions might be right. However, we showed the relative co-localization values of each protein. In particular, the fluorescent intensity in starvation looks lower than control but the relative co-localization signal in starvation was higher than control. As shown in western blot, the substantial amount of SNAI1 might be degraded in starvation, so only 30-40% proteins are available for co-localization while in control 100% protein are available for co-localization. We also examined the relative co-localization between two proteins statistically.

4.     The authors state that there was a large decrease in the levels of mesenchymal proteins including NCAD, Zeb1 and vimentin, which expression was significantly recovered inhibiting autophagy. Looking at Figure 3, the quantification of Zeb1 and Vimentin seems odd and not in line with WB images. I suggest to change the statements (also in discussion) since WB images do not reflect quantification data.

Our response) The statements were modified as suggested. In particular, we modified the statement about the expression of vimentin.

5.     ATG7 knockdown is only showed in combination with starvation treatment, for completeness please add WB of ATG7 siRNA alone. Starvation+siRNA data are convincing, but autophagy induction cannot be appreciated from LC3 and SQSTM1 protein levels, in both cell lines but particularly in H1299 cells, in which LC3 and p62 protein levels seem the same as control, while SNAI1 is downregulated. Can the authors discuss that?

Our response) As suggested by the reviewer, we carried out the additional experiments with ATG7shRNA alone and starvation+ATG7shRNA. New data sets were replaced in the revised manuscript. We also discussed that.

6.     For figure 5 and 6 experiments, the authors focused only on HeLa cell lines, not performing further experiments on H1299. Since the two cell lines do not refer to the same cancer type, can they show if also in H1299 cells they obtained similar results and discuss that?

Our response) We performed all experiments in Figure 5 and 6 using both H1299 and Hela cells. The results were very similar to the ones observed in Hela. We described them in the text of manuscript.

7.    Figure 5D is quite misleading. Even if the downregulation of SNAI1 can be appreciated, the statement that its nuclear translocation is inhibited by autophagy  cannot be corroborated based on these data. The data only show its general degradation, which is not necessarily associated with its nuclear translocation.

Our response) In Figure 5 D we showed that autophagy degrades SNAI1 in the cytoplasm, leading to relative less translocation of SNAI1 to the nucleus. As known, SNAI1, a transcription factor, needs to move into the nucleus for its transcriptional activity. So, we tested the level in cytosol and nucleus as shown in Fig 5. Starvation decreased SNAI1 in cytosol but also in nucleus, while autophagy inhibition with CQ relatively increased the level of SNAI1 in cytosol and also translocation into nucleus as shown in figure 5.    

8.     In the discussion there are several overstatements, as described above, which must be changed. Furthermore, it would be interesting to expand the discussion with previous works that described a similar EMT/Autophagy interplay, like Gugnoni et al (Oncogene. 2017 Feb 2;36(5):667-677. doi: 10.1038/onc.2016.237.), Catalano et al (Mol Oncol. 2015 Oct;9(8):1612-25. doi: 10.1016/j.molonc.2015.04.016.), Lv et al. (Cancer Res. 2012 Jul 1;72(13):3238-50. doi: 10.1158/0008-5472.CAN-11-3832.) or others.

Our response) we fixed some overstatements in the discussion as suggested, and have already discussed some EMT/autophagy links with above references and further expanded  in the revised manuscript.   

Minor comments:

Some minor points are mainly on figures labels (Cq indicated when Bafilomycin is used in fig 6D, an additional “C” to mark the panels in figure 3, Immunohistochemistry instead of immunocytochemistry in paragraph 3.2) and in materials an methods where a lot of drugs and compounds concentrations miss the unit of measure.

Our response) we fixed all as suggested. In Fig 6, we used BafilomycinA1 to inhibit autophagy instead of CQ because CQ were cytotoxic to cells in case of incubation for more than 6 hours. We corrected with concentration and units of drugs in materials and methods. 

Reviewer 2 Report

In this study the authors present a systematic investigation of the role of autophagy in SNAI1 regulation and subsequent epithelial-to-mesenchymal (EMT) transition. They use two cell lines in the study (HeLa and H1299). Is is convincingly shown that induction of autophagy by starvation of treatment with rapamycin results in degradation of SNAI1 and subsequent regulation of its targets.

These are interesting observations and of relevance to the field of autophagy. However, there are a few issues with the study. First the authors have to justify their choice of cell lines. Why HeLa and H1299?

Of major concern are the conclusions drawn from the fluorescence microscopy experiments attempting to determine co-localization of proteins using ImageJ and in Figure 2 C-F: It is not clear how the authors determined the co-localization of SNAI1 and LC3 or SQSTM1. Simple fluorescence microscopy cannot determine for certain that two proteins are co-localized and confocal microscopy is needed. Therefore the authors should exclude the bar graphs in D and F as they are not justified, especially since it is not clear at all from the fluorescent images how the co-localization is determined. The diffuse labeling of SNAI1 makes it impossible to determine co-localization using this technique.

-- The same goes for Fig. 5 D and E. It is not clear how the authors determined nuclear translocation since all the cells seem to have SNAI1 localized in the nucleus, only that the starved cells have a more punctuated nuclear staining.

Minor issues:

Figure 1B and D: there is no need for a legend as it is repetitive of the legend under the X-axis.

Same for figure 4 B and D, Fig. 5, Fig. 6D

Fig. 3: There is a label "C" in the middle of the blots on A.

Author Response

Reviewer2:

In this study the authors present a systematic investigation of the role of autophagy in SNAI1 regulation and subsequent epithelial-to-mesenchymal (EMT) transition. They use two cell lines in the study (HeLa and H1299). Is is convincingly shown that induction of autophagy by starvation of treatment with rapamycin results in degradation of SNAI1 and subsequent regulation of its targets.

These are interesting observations and of relevance to the field of autophagy. However, there are a few issues with the study. First the authors have to justify their choice of cell lines. Why HeLa and H1299?

Our response) We used both HeLa (cervical cancer) and H1299 (lung cancer) cells in experiments because they are well-studied cancer cell model. Since our lab studied autophagy in cancer, we have carried out the experiments and examined many mechanistic aspects of autophagy using these cell lines.

Of major concern are the conclusions drawn from the fluorescence microscopy experiments attempting to determine co-localization of proteins using ImageJ and in Figure 2 C-F: It is not clear how the authors determined the co-localization of SNAI1 and LC3 or SQSTM1. Simple fluorescence microscopy cannot determine for certain that two proteins are co-localized and confocal microscopy is needed. Therefore the authors should exclude the bar graphs in D and F as they are not justified, especially since it is not clear at all from the fluorescent images how the co-localization is determined. The diffuse labeling of SNAI1 makes it impossible to determine co-localization using this technique.

Our response) There are many software including ImageJ to quantify the fluorescent images. We used the BX51-DSU fluorescent microscope to obtain high quality images, very compatible to images acquired from the confocal microscope. We have already published some co-localization data in high-impact journals (e.g. Autophagy, Cancers) using ImageJ with co-localization Plug-in or our newly developed R-package (under preparation for publication). We also analyzed image data as described by The art RP, Loos B, Powrie YSL, Niesler TR (2018) Improved region of interest selection and colocalization analysis in three-dimensional fluorescence microscopy samples using virtual reality. PLoS ONE 13(8): e0201965.

The same goes for Fig. 5 D and E. It is not clear how the authors determined nuclear translocation since all the cells seem to have SNAI1 localized in the nucleus, only that the starved cells have a more punctuated nuclear staining.

Our response) In Figure 5 D we showed that autophagy degrades SNAI1 in the cytoplasm, leading to relative less translocation of SNAI1 to the nucleus. As known, SNAI1, a transcription factor, needs to move into the nucleus for its transcriptional activity. So, we tested the level in cytosol and nucleus as shown in Fig 5. Starvation decreased SNAI1 in cytosol but also in nucleus, while autophagy inhibition with CQ relatively increased the level of SNAI1 in cytosol and also translocation into nucleus as shown in figure 5.

Minor issues:

Figure 1B and D: there is no need for a legend as it is repetitive of the legend under the X-axis. Same for figure 4 B and D, Fig. 5, Fig. 6D. Fig. 3: There is a label "C" in the middle of the blots on A.

Our response) According to the reviewer suggestion, all x- axis legends have been removed in all figures. The label “C” has been removed.

Round 2

Reviewer 1 Report

The Authors only partially addressed the concerns that were raised on the previous version. 

In particular two major points remain:

SNAI1 immunofluorescence (Figure 2 and Figure 5). , SNAI signaling is largely diffuse and the images presented cannot be used to estimate quantitive localization, independently from the software used. This point was raised also by the other Review and in my opinion the Authors answer was not appropriate and did not resolved this issue.

I did not understand the point about H1299 experiments. I did not find the figures or any reference to this in the text. If this is not dependent by me missing something and the Authors have performed the experiments in H1299 please provide prove of them. 

Finally, the Authors claim they have reduced overstating about the interpretation of their results. However, I don't feel this has been done sufficiently and the discussion still presents conclusion that are supported by the data.

Reviewer 2 Report

The authors should include a brief paragraph justifying their method of co-localizing proteins using fluorescence microscopy that is not confocal microscopy.